# Robust Alzheimer's Progression Modeling using Cross-Domain Self-Supervised Deep Learning

**Saba Dadsetan** *dadsetas@gene.com*
*Intelligent Systems Program, University of Pittsburgh, Pittsburgh, PA, USA*
*Department of Artificial Intelligence, Genentech Inc., South San Francisco, CA, USA*

**Mohsen Hejrati** *hejratis@gene.com*
*Department of Artificial Intelligence, Genentech Inc., South San Francisco, CA, USA*

**Shandong Wu** *shw83@pitt.edu*
*Intelligent Systems Program, University of Pittsburgh, Pittsburgh, PA, USA*

**Somaye Hashemifar** *hashemifar.somaye@gene.com*
*Department of Artificial Intelligence, Genentech Inc., South San Francisco, CA, USA*

**for the Swedish BioFINDER study group**[*]

**The Alzheimer's Disease Neuroimaging Initiative**[†]

**for the BIOCARD Study team**[‡]

**Reviewed on OpenReview:** `https://openreview.net/forum?id=HVAeM6sNo8`

## Abstract

Developing successful artificial intelligence systems in practice depends on both robust deep learning models and large, high-quality data. However, acquiring and labeling data can be prohibitively expensive and time-consuming in many real-world applications, such as clinical disease models. Self-supervised learning has demonstrated great potential in increasing model accuracy and robustness in small data regimes. In addition, many clinical imaging and disease modeling applications rely heavily on regression of continuous quantities. However, the applicability of self-supervised learning for these medical-imaging regression tasks has not been extensively studied. In this study, we develop a cross-domain self-supervised learning approach for disease prognostic modeling as a regression problem using medical images as input. We demonstrate that self-supervised pretraining can improve the prediction of Alzheimer's Disease progression from brain MRI. We also show that pretraining on extended (but not labeled) brain MRI data outperforms pretraining on natural images. We further observe that the highest performance is achieved when both natural images and extended brain-MRI data are used for pretraining.

---

[*]A complete list of the BioFINDER study group members can be found at `www.biofinder.se`.

[†]Data used in preparation of this article were obtained from the Alzheimer's Disease Neuroimaging Initiative (ADNI) database (adni.loni.usc.edu). As such, the investigators within the ADNI contributed to the design and implementation of ADNI and/or provided data but did not participate in analysis or writing of this report. A complete listing of ADNI investigators can be found at `http://adni.loni.usc.edu/wpcontent/uploads/how_to_apply/ADNI_Acknowledgement_List.pdf`

[‡]Data used in preparation of this article were derived from BIOCARD study, supported by grant $U19 - AG033655$ from the National Institute on Aging. The BIOCARD study team did not participate in the analysis or writing of this report, however, they contributed to the design and implementation of the study. A listing of BIOCARD investigators can be found on the BIOCARD website (on the 'BIOCARD Data Access Procedures' page, 'Acknowledgement Agreement' document).

# 1 Introduction

Developing reliable and robust artificial intelligence systems requires efficient deep learning techniques, as well as the creation and annotation of large volumes of data for training. However, the construction of a labeled dataset is often time-consuming and expensive, such as in the medical imaging domain, due to the complexity of annotation tasks and the high expertise required for the manual interpretation. To alleviate the lack of annotations in medical imaging, transfer learning from natural images is becoming increasingly popular (Liu et al., 2020b; McKinney et al., 2020; Menegola et al., 2017; Xie et al., 2019). Although numerous experimental studies indicate the effectiveness of fine-tuning from either supervised or self-supervised ImageNet models( (Alzubaidi et al., 2020; Graziani et al., 2019; Heker & Greenspan, 2020; Zhou et al., 2021; Hosseinzadeh Taher et al., 2021; Azizi et al., 2021)), it does not always improve the performance due to domain mismatch problem (Raghu et al., 2019).

In the meantime, self-supervised learning (SSL) has demonstrated great success in many downstream applications in computer vision, where the labeling process is quite expensive and time-consuming (Doersch et al., 2015; Gidaris et al., 2018; Noroozi & Favaro, 2016; Zhang et al., 2016; Ye et al., 2019; Bachman et al., 2019; Tian et al., 2020a; Henaff, 2020; Oord et al., 2018). Medical research and healthcare are especially well-poised to benefit from SSL learning approaches, given the prevalence of unprecedented amounts of medical images generated by hospital and non-hospital settings. Despite this demand, the use of SSL approaches in the medical image domain has received limited attention. Only a few studies have investigated the impact of SSL in the medical image analysis domain for limited applications including classification (Liu et al., 2019; Sowrirajan et al., 2021; He et al., 2020b; Azizi et al., 2022; Zhu et al., 2020; Liu et al., 2020a) and segmentation (Ronneberger et al., 2015; Bai et al., 2019; Chaitanya et al., 2020; Spitzer et al., 2018).

In this study, we focus on developing a self-supervised deep learning model for predicting the progression of Alzheimer's disease (AD) by using high dimensional magnetic resonance imaging (MRI). AD is a slowly progressing disease caused by the degeneration of brain cells, with patients showing clinical symptoms years after the onset of the disease. Therefore, accurate prognosis and treatment of AD in its early stage is critical to prevent non-reversible and fatal brain damage. By accurately predicting the progression of AD, clinicians can start treatment earlier and provide more personalized care.

Many existing methods for predicting AD progression have focused on classifying patients into coarse categories, such as Mild Cognitive Impairment (MCI) or dementia, as well as predicting conversion from one category to another (e.g. MCI to dementia) (Risacher et al., 2009; Venugopalan et al., 2021; Oh et al., 2019). However, applications in clinical trials require a more fine-grained measurement scale, because clinical trial populations are typically narrowly defined (eg. only MCI). An alternative approach is to predict the outcome of cognitive and functional tests, which are represented by continuous numerical values. By framing the prediction task as a regression rather than a classification, prognostic models offer more granular estimates of disease progression. Recently, a few deep learning based approaches, including the recurrent neural network (RNN) and convolutional neural networks (CNN) have been proposed for predicting disease progression of AD patients based on MRIs. Nguyen et al. (2020) adapted MinimalRNN to integrate longitudinal clinical information and cross-sectional tabular imaging features for regressing endpoints. El-Sappagh et al. (2020) utilized an ensemble model based on stacked CNN and a bidirectional long short-term memory (BiLSTM) to predict the endpoints on the fusion of time series clinical features and derived imaging features. Recently, several methods have started to employ CNN based models to extract features from raw medical imaging. Tian et al. (2022) applied CNN with a multi-task interaction layers composed of feature decoupling modules and feature interaction module to predict the disease progression.

Despite demand, little progress has been made because of the difficult design requirements, lack of large-scale, homogeneous datasets that contain early stage AD patients, and noisy endpoints that are potentially hard to predict. Much of the prior work has focused on using image-derived features to overcome the complexity and high variability in raw MRIs, and small datasets. Most current prognosis models are trained on a single dataset (i.e. cohort), which limits their generalizability to other cohorts. They also use a limited number of annotated images, which can lead to problems such as domain shift and heterogeneity.

We established a cross-domain self-supervised transfer learning approach that learns transferable and generalizable representations for medical images. Our approach leverages SSL on both unlabeled large-scale natural images and an in-domain medical image dataset comprised from 11 different internal and external studies. These representations can be further fine-tuned for downstream tasks such as disease progression prediction, with using limited labeled data from the clinical setting. We evaluated the performance of different supervised and self-supervised models pretrained on either natural images or medical images, or both. Our extensive experiments reveal that (1) Self-supervised pretraining on natural images followed by self-supervised learning on unlabeled medical images outperforms alternative transfer learning methods, indicating the potential of SSL in reducing the reliance on data annotation compared to supervised approaches (2) Self-supervised models pretrained on medical images outperform those pretrained on natural images, denoting that SSL on medical images yield discriminative feature representations for regression task.

## 2 Related works

The recent development and success of self-supervised learning techniques, including contrastive learning (Wu et al., 2018; He et al., 2020a; Chen et al., 2020c;b;a; Grill et al., 2020; Misra & Maaten, 2020), mutual information reduction (Tian et al., 2020b), clustering (Caron et al., 2020; Li et al., 2020), and redundancy-reduction methods (Zbontar et al., 2021; Bardes et al., 2021) in computer vision indicate their effectiveness in improving the performance of AI systems. These methods train models on different pretext tasks to enable the network to learn high-quality representations without label information. SimCLR (Chen et al., 2020c) maximizes agreement between representations of different augmentations of the same image by using a contrastive loss in the latent space. Barlow Twins (Zbontar et al., 2021) measure the cross-correlation matrix between the embedding of two identical networks and its goal is to make this cross-correlation close to the identity matrix. SwAV (Caron et al., 2020) simultaneously clusters the images while enforcing consistency between cluster assignments produced for differently augmented views of the same image, instead of comparing features directly as in contrastive learning.

Subsequently, self-supervised learning has been employed for medical imaging applications including classification and segmentation to learn visual representations of medical images by incorporating unlabeled medical images. While some approaches have designed domain-specific pretext tasks (Bai et al., 2019; Spitzer et al., 2018; Zhuang et al., 2019; Zhu et al., 2020), others have adjusted well-known self-supervised learning methods to medical data (He et al., 2020b; Li et al., 2021; Zhou et al., 2020; Sowrirajan et al., 2021). Very recently Azizi et al. (2022) has applied SimCLR on a combination of unlabeled ImageNet dataset and task specific medical images for medical image classification; their experiments and improved performance suggest that pretraining on ImageNet is complementary to pretraining on unlabeled medical images.

Although aforementioned approaches demonstrate improvement of the performance on challenging medical datasets, all of them are limited to classification and segmentation tasks and their benefits and potential effects for the prognosis prediction tasks, as regression tasks, have not been studied. Formulating progress prediction as a regression rather than a traditional classification problem leads to a more fine-grained measurement scale which is crucial for real-world applications. Therefore, the development of self-supervised networks is in great demand for efficient data-utilization in medical imaging for disease prognosis. To the best of our knowledge, this is the first study of developing a self-supervised deep convolution neural network on medical data images from various cross-domain datasets to predict a granular understanding of disease progression.

## 3 Methodology

### 3.1 Task and Data

In Alzheimer's Disease clinical trials, the most important cognitive test that is performed to assess current patient function and the likelihood of AD progression is Clinical Dementia Rating Scale Sum of Boxes (CDR-SB). CDR-SB is a score provided by clinicians based on clinical evaluations and its ranges from 0 to 18, with higher scores indicating greater severity of symptoms. CDR-SB score is then used to assign Alzheimer's status of a patient.

Our goal is to use a regression approach to predict the future status of patients with AD based on their initial visit. Specifically, our model takes as input 2D slice stacks of an MRI volume that are collected at the first visit and predicts the CDR-SB value at month 12 (i.e. after one year). By accurately predicting the progression of AD in patients, clinicians can initiate treatment at an earlier stage and tailor the most suitable and effective treatment for each individual patient.

All individuals included in our analysis are around $10k$ from eight external studies, including ADNI (Petersen et al., 2010), BIOFINDER (Mattsson-Carlgren et al., 2020), FACEHBI (Moreno-Grau et al., 2018), AIBL (Ellis et al., 2009), HABS (Dagley et al., 2017), BIOCARD (Moghekar et al., 2013), WRAP (Langhough Koscik et al., 2021), and OASIS-3 (LaMontagne et al., 2019), and five internal studies including Scarlet RoAD (NCT01224106), Marguerite RoAD (NCT02051608), CREAD 1 & 2 (NCT02670083, NCT03114657), BLAZE (NCT01397578), and ABBY (NCT01343966).

MR volumes were standardized using the following preprocessing steps. First, a brain mask was inferred for each volume using SynthSeg (Billot et al., 2021), a deep learning segmentation package. During training, the volumes and segmentations were resampled isotropically to 1 mm voxel size, standardized to canonical (RAS+) orientation, intensity rescaled to 0,1 and Z-score normalized. Finally, volumes are cropped or padded to (224,224).

For training self-supervised models, we offer two training sets of medical images called the center-slice and 5-slice datasets. The 5-slice dataset is generated by extracting a stack of five slices from the 3D MRI volumes. This stack comprises the center slice of the brain sub-volumes and four adjacent slices, with intervals of 5 (two to the right, two to the left). In contrast, the center-slice dataset only includes the center slice. Both training sets include subjects from all datasets except OASIS-3 and dataset ABBY , which are reserved as out-study test sets (see Table 1). The 5-slice dataset contains a total of $218,700$ unlabeled images, while the center-slice dataset comprises $43,740$ unlabeled images. Around 90% of the development set is used for training, while the remaining 10% is set aside for validation. We select the best model based on the minimum self-supervised validation loss and transfer its backbone weights to the supervised model.

To create a labeled datasets for fine tuning, the participants are selected from five studies, including ADNI, Scarlet RoAD, Marguerite RoAD, CREAD 1 & 2, and BLAZE. All studies were filtered to include patients who are positive for amyloid pathology, have a MMSE larger than 20 and are diagnosed with early stages of AD i.e. prodromal or mild at baseline. Amyloid positivity is detected by either cerebrospinal fluid (CSF) or amyloid positron emission tomography (PET). The fine-tuning dataset consists of approximately 1000 images, none of which are used for self-supervised learning training. Roughly 30% of the fine-tuning dataset is reserved as an in-study test set, while the remaining data is divided into training and validation sets (see Table 1).

We have three test sets: one in-study test set and two out-study test sets. The in-study test set consists of one-third of the patients from ADNI, Scarlet RoAD, Marguerite RoAD, CREAD 1 & 2, and BLAZE. The two out-study test sets are OASIS-3 and study ABBY, that are not utilized for either self-supervised pretraining or fine-tuning. The primary purpose of these out-study test sets is to assess the generalizability of our model on external datasets.

### 3.2 Self-supervised learning platform for progression prediction task

We evaluated the performance of three SSL pretraining approaches in predicting disease progression. The first approach was to explore the pretrained models on unlabeled natural images to see if they could be transferred to medical images. The second approach was to use pretrained models on unlabeled in-domain medical images to assess their performance on disease progression. The third approach was to apply cross-domain SSL (referred to as CDSSL) to leverage unlabeled data from multiple domains, including natural images and medical images. To establish a reference point, the target model is trained using random initialization, serving as a baseline for comparison.

**Exploring pretrained models on unlabeled natural images**

SSL models trained on large datasets of natural images, such as ImageNet, have been shown to outperform supervised ImageNet models on several computer vision tasks. In this experiment, we hypothesized that

| Study | Number of patients | SSL | Fine-tuning | in-study test | out-study test |
|---|---|---|---|---|---|
| HABS | 289 | ✓ | - | - | - |
| BIOFINDER | 752 | ✓ | - | - | - |
| BIOCARD | 434 | ✓ | - | - | - |
| AIBL | 1112 | ✓ | - | - | - |
| WRAP | 578 | ✓ | - | - | - |
| FACEHBI | 236 | ✓ | - | - | - |
| ADNI | 2332 | ✓ | ✓ | ✓ | - |
| Scarlet RoAD | 797 | ✓ | ✓ | ✓ | - |
| Marguerite RoAD | 470 | ✓ | ✓ | ✓ | - |
| CREAD 1 & 2 | 2417 | ✓ | ✓ | ✓ | - |
| BLAZE | 91 | ✓ | ✓ | ✓ | - |
| ABBY | 445 | - | - | - | ✓ |
| OASIS-3 | 46 | - | - | - | ✓ |

Table 1: Summary of datasets. The ✓ indicates whether a study is utilized for a split. OASIS-3 and study ABBY are designated as out-study test sets, meaning they have not been utilized for either SSL pretraining or fine-tuning. The in-study test set includes patients from ADNI, Scarlet RoAD, Marguerite RoAD, CREAD 1 & 2, and BLAZE; but there is no overlap between the splits. We were unable to find any labels for the first 6 rows of datasets

these models could be transferred to medical images and used to predict disease progression. We initialized the backbone encoder with weights from SSL models trained on ImageNet to exploit these benefits. In our study, we specifically concentrate on three prominent SSL methods: SimCLR, a contrastive approach; BarLow Twins (BLT), a redundancy reduction approach; and SwAV, a clustering-based contrastive learning approach. These methods have demonstrated remarkable performance on benchmarks designed for natural images. Although there are other SSL strategies available, their performance on ImageNet is comparable to the ones we have selected.

**Exploring pretrained models on unlabeled in-domain medical images**

As shown in Figure 1(a), we used SimCLR, Barlow Twins, and SwAV to learn distinctive representations of unlabeled medical images. These methods have all been shown to be effective in the classification and segmentation of medical images. We hypothesized that these models would be able to learn the features that are specific to medical images and be more effective at predicting disease progression than the models trained on unlabeled natural images.

**Exploring CDSSL pretrained models on both domains**

Representations learned from natural images may not be optimal for the medical imaging domain because of the large distribution shift between natural and medical images. Medical images are typically monochromatic and have similar anatomical structures, while natural images are typically colorful and have a wider variety of objects and scenes. We hypothesized that this discrepancy could be minimized by further pretraining on medical data. As shown in Figure 1(b-c), we used SimCLR, Barlow Twins, and SwAV to learn distinctive representations of unlabeled medical images on top of pretraining on ImageNet .

**Fine Tuning**

The progression prediction task utilizes a ResNet50 backbone (He et al., 2016) followed by a linear layer, with the backbone being initialized randomly or with pretrained models. The model loss is calculated using the mean square error (MSE) criterion (see Figure 1(d)).

## 4 Results

We proposed the first benchmarking study to evaluate the effectiveness of different pretraining models for disease progression prediction as a regression problem. Our main objective is to investigate the transferability

of features learned by pretraining on natural or medical images, or both, to the medical task of disease progression prediction. To evaluate our model's performance, we used the Pearson correlation coefficient ($r$) and the coefficient of determination ($R^2$). $R^2$ is calculated as bellow:

$$R^2 = 1 - \frac{\sum (y_i - \hat{y})^2}{\sum (y_i - \bar{y})^2} \tag{1}$$

where $\sum (e_i^2)$ and $\sum (y_i - \bar{y})^2$ respectively indicate the sum of squared residuals and total sum squared. In clinical studies, $R^2$ is widely employed to evaluate the ability of a model to predict future outcomes (Franzmeier et al., 2020).

## 4.1 Self-Supervised Pretraining

**Experiments On Natural Images.** This experiment examined the transferability of standard ImageNet models through the utilization of three widely recognized SSL methods: SimCLR, Barlow Twins, and SwAV. All SSL models undergo pretraining on the ImageNet dataset and employ a ResNet-50 backbone.

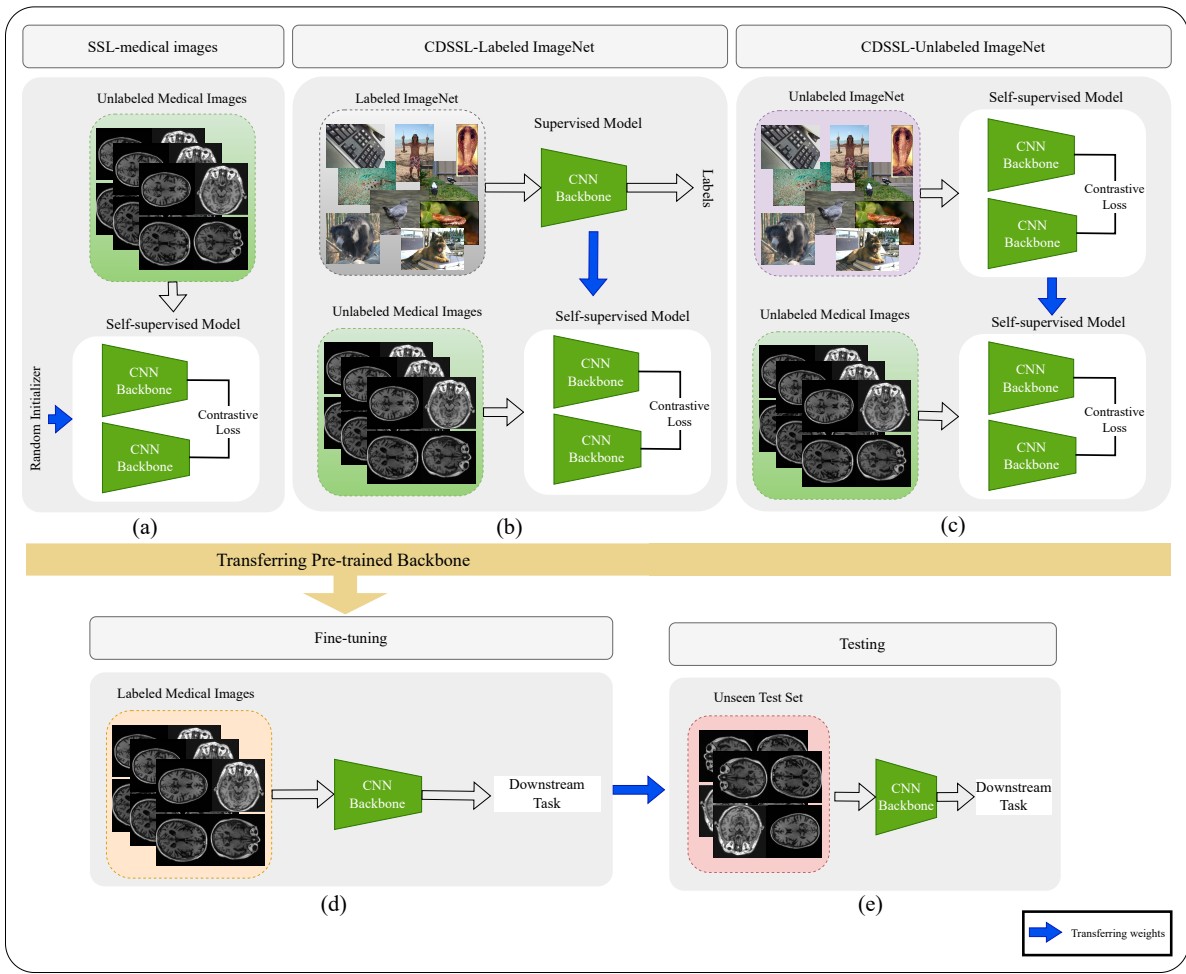

Figure 1: Different approaches for self-supervised pretraining on in-domain medical imaging, including (a) random initialization, (b) supervised ImageNet initialization, and (c) self-supervised ImageNet initialization. (d) Performing fine-tuning by transferring the backbone from one of the scenarios a-c. (e) utilization of the trained model on unseen test sets.

| pretraining | Initialization | $R^2$ |
|---|---|---|
| - | Random | 0.07 |
| Supervised | ImageNet | 0.06 |
| | SimCLR | 0.10 |
| Self-Supervised | SwAV | 0.08 |
| | Barlow Twins | **0.14** |

(a) pretraining on Natural Images.

| pretraining | Initialization | $R^2$ |
|---|---|---|
| - | Random | 0.07 |
| | SimCLR | 0.17 |
| Self-Supervised | SwAV | 0.10 |
| | Barlow Twins | 0.13 |

(b) pretraining on Medical Images.

Table 2: Comparison of different pretraining schemes on downstream task.

**Results.** The results in Table 2a show that transfer learning from the supervised ImageNet model does not improve over random initialization. This is likely due to the significant domain shift between the pretraining and regression target task. In particular, supervised ImageNet models tend to capture domain-specific semantic features, which can be inefficient if the pretraining and target data distributions differ. This finding is consistent with recent studies on other medical tasks that demonstrate that transfer learning from supervised ImageNet pretraining does not always correlate with performance on either classification or segmentation of medical images (Dippel et al., 2021; Vendrow & Schonfeld, 2022; Hosseinzadeh Taher et al., 2021).

In contrast, transfer learning from self-supervised ImageNet models provides superior performance compared with both random initialization and transfer learning from the supervised ImageNet model. The best self-supervised model (i.e., Barlow Twins) achieves a performance improvement of 7% and 8% over random initialization and the supervised ImageNet model, respectively. This is likely due to the fact that self-supervised ImageNet models are trained to learn general-purpose features that are not biased toward any particular task. As a result, they are better able to generalize to new domains.

**Experiments On Medical Images.** To investigate the effect of using in-domain medical images for self-supervised pretraining, we trained three SSL methods, SimCLR, Barlow Twins, and SwAV, on 11 unlabeled medical imaging datasets, which we call in-domain datasets. All SSL models were randomly initialized and then fine-tuned on our labeled dataset.

**Results.** Table 2b shows the performance of SSL models pretrained on the center slice dataset, measured by the $R^2$ score. We observe that SimCLR pretraining on the in-domain dataset achieves the highest performance, providing a 7% and 4% boost over SwAV and Barlow Twins, respectively. This may be due to the superiority of contrastive learning for identifying significant MRI features for predicting progression of Alzheimer's disease in terms of CDR-SB. Moreover, the performance of SimCLR pretraining on the in-domain dataset exceeds that of both supervised and self-supervised pretraining on the ImageNet dataset (as seen in Table 2a). This suggests that pretraining on the in-domain dataset encodes domain-specific features that reflect the distinctive characteristics of medical images.

In contrast, pretraining Barlow Twins on in-domain data does not yield performance improvement compared to Barlow Twins pretrained on ImageNet. This result indicates that the features learned by Barlow Twins through pretraining on ImageNet demonstrate sufficient generalizability to medical images. Thus, the limited number of unlabeled medical images in the in-domain dataset (40k compared to 1.3M in ImageNet) may only provide marginal performance gains for the redundancy reductions-based Barlow Twins method.

**Scaling Ablations.** We conducted further experiments to evaluate the benefits of using more unlabeled images for self-supervised pretraining. For each SSL method, we trained two separate models, one pretrained on center-slice dataset and the other pretrained on 5-slice dataset.

**Results.** As shown in Table 3, all three SSL models that were pretrained on the 5-slice dataset achieved better results than those that were pretrained on the center-slice dataset. This suggests that SSL models can benefit from larger in-domain unlabeled datasets for pretraining. Specifically, the performance of Barlow Twins improved by 1% when it was pretrained on the 5-slice dataset. This may confirm our previous assumption that Barlow Twins, in contrast to SimCLR, requires more unlabeled in-domain images to constrain the ImageNet-based features for medical tasks.

| Self-Supervised Model | Dataset | $R^2$ |
|---|---|---|
| SimCLR | center-slice | 0.17 |
| | 5-slice | **0.19** |
| SwAV | center-slice | 0.10 |
| | 5-slice | 0.12 |
| Barlow Twins | center-slice | 0.13 |
| | 5-slice | 0.14 |

Table 3: Performance comparison of self-supervised models pretrained on dataset of different sizes

**Cross-Domain Experiments.** In this experiment, we examine the effects of self-supervised pretraining on both natural images and medical images. To achieve this, we pretrained SimCLR on the 5-slice dataset with two distinct initialization schemes: Supervised ImageNet (referred to as Labeled_ImageNet→In-domain), and Barlow Twins on the ImageNet dataset (referred to as Unlabeled_ImageNet→In-domain). We selected SimCLR and Barlow Twins for our experiments because they achieved the highest performance when pretrained on either natural or medical images, respectively, as shown in Tables 2a and 2b. Figure 1(b,c) shows both cross-domain self-supervised pretraining schemes. In this section, we also include the Pearson correlation coefficients to provide insights into the strength and direction of the relationship between the predicted values and the target values of CDR-SB.

**Results.** The results are displayed in Table 4a. It is evident that the highest performance is achieved when both the unlabeled ImageNet and in-domain datasets are utilized for pretraining. Specifically, the Unlabeled_ImageNet→In-domain pretraining surpasses the performance of models pretrained solely on the in-domain dataset or ImageNet. It yields significant performance improvements of 14%, 7%, and 2% compared to random initialization, ImageNet pretraining alone, and in-domain dataset pretraining alone, respectively. Consistent with earlier research (Hosseinzadeh Taher et al., 2021; Azizi et al., 2021), these findings suggest that combining pretraining on ImageNet with pretraining on in-domain datasets leads to more robust representations for medical applications. Additionally, it is worth mentioning that the Labeled_ImageNet→In-domain pretraining approach demonstrates inferior performance compared to the Unlabeled_ImageNet→In-domain approach. These observations restate the effectiveness of self-supervised models in generating more generic representations that can be applied to target tasks with limited data, thereby reducing the need for extensive annotations and associated costs.

To compare the Pearson correlation coefficients of the models in Table 4a, we used Steiger's Z1 method (Steiger, 1980). This method is utilized to compute the two-tailed p-value at a 95% confidence interval, and it's a statistical approach uniquely suited for assessing the significance of variances between dependent correlation coefficients. The results are displayed in Table 5. Notably, the Unlabeled_ImageNet→In-domain pretraining model demonstrates a statistically significant difference compared to other models.

Figure 2 illustrates the average frequency distribution of residuals across all three folds for the top-performing models. Mean and standard deviation values are provided for each plot. Notably, the cross-domain model, specifically Barlow Twins→SimCLR, exhibits a higher concentration of residuals near zero and demonstrates smaller mean and standard deviation values.

### 4.2 In- and out-of-Domain Generalization

To assess the robustness of our top-performing model, Barlow Twins→SimCLR, on the 5-slice dataset, we evaluated its performance using three distinct test sets: an in-study test set and two out-study test sets. Furthermore, we compare this model's performance with the best-performing models initialized either by ImageNet or an in-domain dataset.

According to the results presented in Table 4b and Table 6, the highest performance is achieved when both the unlabeled ImageNet and in-domain dataset are utilized for pretraining. Specifically, the Unlabeled_ImageNet→In-domain pretraining approach exhibits a significant improvement of 10% and 6%

over the in-domain and ImageNet pretrained models, respectively, for the in-study test set. Similarly, on out-study test sets ABBY and OASIS-3, the same model demonstrates an improvement up to 6% compared to other models. These results indicate that Unlabeled_ImageNet→In-domain pretraining effectively encodes semantic features that are generalizable to other studies. Furthermore, the performance of Labeled_ImageNet→In-domain pretraining is inferior to that of Unlabeled_ImageNet→In-domain pretraining. This indicates that supervised ImageNet models encode domain-specific semantic features, which may not be efficient when the pretraining and target data distributions significantly differ.

## 4.3 Visualizing Model Saliency Maps

Attribution methods are a tool for investigating and validating machine learning models. Using the interpretability of the ML models, can significantly help obtaining a bigger picture about risk factors influences on short-term prognosis. We used the GradCAM (Selvaraju et al., 2017) method to extract and evaluate the varying importance of each part of brain MRIs using a gradient of final score. In this method, regions of an image are marked with different colors ranging from red to blue. Generally, areas that are closer to the red color contribute more significantly to the final result, based on the input data (i.e., MRI slice).

| Pretraining Method | Pretraining Dataset | $R^2$ | $r$ | $MSE$ |
|---|---|---|---|---|
| Random | - | 0.07 (0.03) | 0.33 (0.04) | 5.31 (0.68) |
| Barlow Twins | ImageNet | 0.16 (0.01) | 0.42 (0.01) | 4.81 (0.65) |
| SimCLR | In-domain | 0.19 (0.02) | 0.44 (0.01) | 4.61 (0.62) |
| Supervised ImageNet → SimCLR | Labeled_ImageNet → In-domain | 0.17 (0.04) | 0.43 (0.05) | 4.74 (0.82) |
| Barlow Twins → SimCLR | Unlabeled_ImageNet → In-domain | **0.21** (0.02) | **0.46** (0.01) | **4.52** (0.56) |

(a) Results of best performing models in each domain and their combination in cross-domain SSL setting on validation set. All values are the mean over 3-fold stratified cross-validation, reported with standard deviation in brackets

| Pretraining Method | Pretraining Dataset | $R^2/r$ on in-study test | $R^2/r$ on ABBY | $R^2/r$ on OASIS-3 |
|---|---|---|---|---|
| Random | - | 0.04/0.30 | 0.02/0.29 | -0.04/0.10 |
| Barlow Twins | ImageNet | 0.12/0.35 | 0.07/0.34 | -0.06/0.15 |
| SimCLR | In-domain | 0.14/0.38 | 0.09/0.35 | 0.11/0.36 |
| Supervised ImageNet → SimCLR | Labeled_ImageNet → In-domain | 0.11/0.33 | 0.11/0.35 | 0.10/0.35 |
| Barlow Twins → SimCLR | Unlabeled_ImageNet → In-domain | **0.18/0.42** | **0.14/0.38** | **0.17/0.42** |

(b) Results on independent dataset

Table 4: Results of different pretraining schemes on both (a) validation and (b) test sets in terms of $R^2$ and $r$. OASIS-3 and ABBY are out-study test sets, meaning they have not been utilized for either SSL pretraining or fine-tuning.

| Pretraining Method | Random | Barlow Twins | SimCLR | Supervised ImageNet → SimCLR | Barlow Twins → SimCLR |
|---|---|---|---|---|---|
| Random | - | 0.003** | 0.0001*** | 0.0001*** | 0.0001*** |
| Barlow Twins | 0.003** | - | 0.04* | 0.27 | 0.0004*** |
| SimCLR | 0.0001*** | 0.04* | - | 0.35 | 0.012* |
| Supervised ImageNet → SimCLR | 0.0001*** | 0.27 | 0.35 | - | 0.004** |
| Barlow Twins → SimCLR | 0.0001*** | 0.0004*** | 0.012* | 0.004* | - |

Table 5: Statistical significance of the resulting Pearson correlation coefficients. The p-values indicate the statistical significance of difference between the Pearson correlation coefficients of different models in Table4a. Significance levels indicated by *, **, and *** for p-values $< 0.05$, $< 0.01$, and $< 0.001$, respectively.

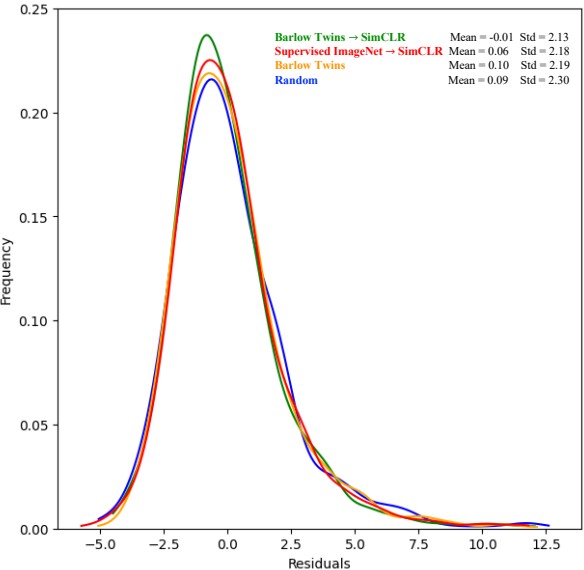

Figure 2: Average frequency distribution. Each color represents the frequency distribution of residuals for a specific experiment.

Figure 3 presents various examples of saliency maps, which depict the significance of different regions in the MRIs at a pixel level. Notably, our top-performing model, Barlow Twins→SimCLR, consistently highlights the subcortical areas of the brain. This observation aligns with prior research indicating that the initial stages of AD exhibit abnormal tau accumulation in the entorhinal cortex and subcortical brain regions (Rueb et al., 2017; Liu et al., 2012). Therefore, it is reasonable that our model, i.e. Barlow Twins→SimCLR, exhibits higher attention to those specific brain regions in a population of individuals with prodromal to mild Alzheimer's disease at baseline.

The randomly initialized model highlights random features all around MRIs, including background areas. In contrast, the model initialized with unsupervised ImageNet is more focused on brain regions, rather than irrelevant areas such as the background.

## 5 Discussion

We present a cross-domain self-supervised learning framework for predicting the progression of Alzheimer's disease from MRIs, which is formulated as a regression task. Our study is a pioneering effort to aggregate a comprehensive set of internal and external cohorts/studies to create a substantial dataset for model training. we show that using SimCLR pretraining on natural images followed by pretraining on medical images achieved the highest accuracy. This approach effectively alleviates domain shift challenge and greatly improves the generalization of the pretrained features.

Our extensive experiments demonstrate the effectiveness of our approach to combat the lack of large-scale annotated data for training deep models for progression prediction. Our best performing model exhibits a substantial improvement over the fully supervised model, demonstrating that the appropriate utilization of unlabeled images, including both natural and medical images, indeed provides additional useful information that the model successfully learns from. Moreover, it indicates that the proposed Cross-Domain self supervised learning approach can learn domain-invariant features, which enhances model generalization ability and robustness. Therefore, it has the potential to identify patients at higher risk of progressing to AD and help develop better therapies at lower cost to society.

Our model has limitations related to the heterogeneity of the datasets used to train it, including differences in criteria for amyloid positivity across studies. Because of limited resources, our dataset analysis is focused

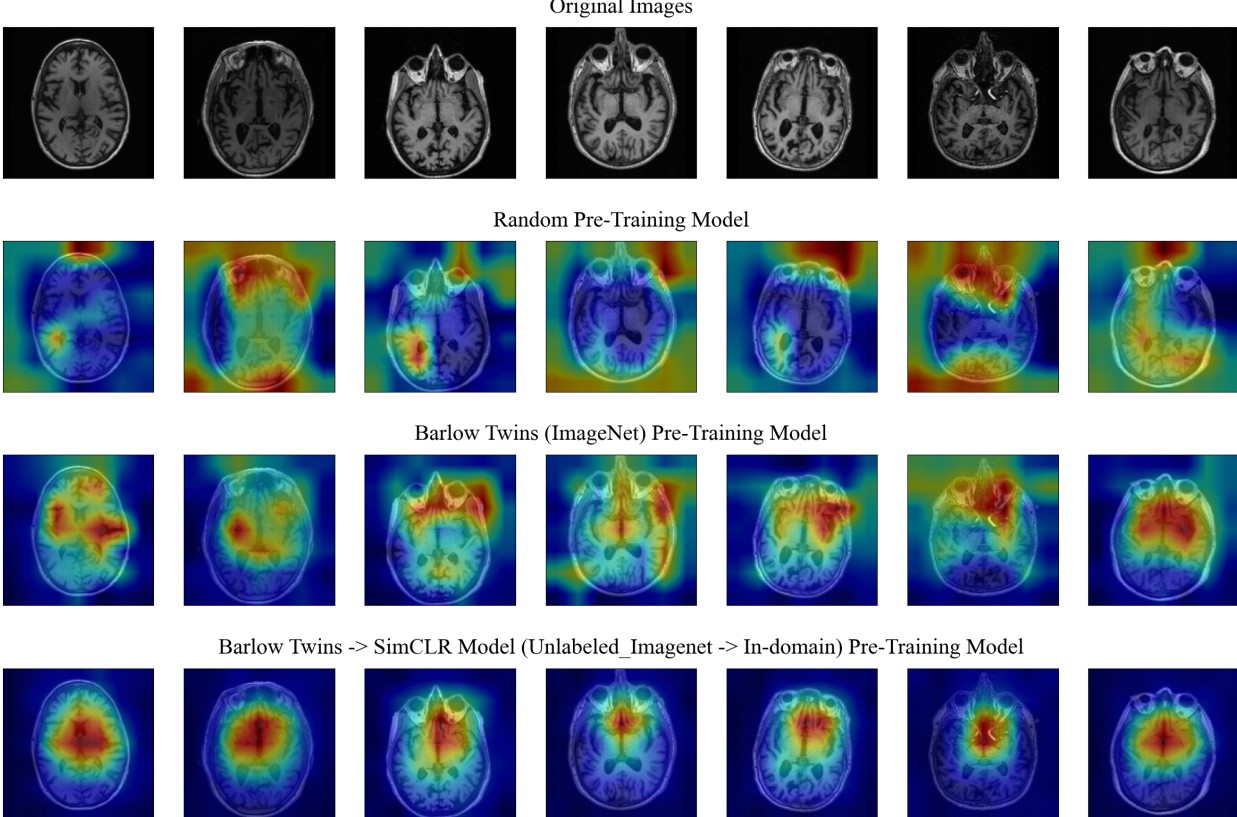

Figure 3: Illustrating the interpretation of three pretraining models using GradCam technique. The top row showcases the original MRI slices, while the subsequent rows, from top to bottom, illustrate the saliency maps generated by the following models: a randomly-initialized pretrained model, a pretrained model on natural images, and our best model, Barlow Twins → SimCLR.

on a subset of five slices per MRI scan. As a result, we are only able to utilize a small portion of the MRI volumes information in each scan. It is important to note that this constraint may hinder our model's ability to effectively learn from the complete set of brain regions. These limitations should be taken into consideration when interpreting the results from our model.

Our research only explores the benefits of 2D MRIs for prognostic prediction. This reduces the number of parameters and simplifies the model. 3D deep neural networks require many more learnable parameters to solve such a complex problem, and they do not always produce accurate results (Wang et al., 2019). In future work, we will expand our dataset to include more than 5 slices from each image, as well as other views such as coronals and sagittals, allowing us to introduce more in-domain information to our pretraining model. We also will incorporate 3D MRIs into our analysis to compare their performance against 2D slices.

**Acknowledgements**

We would like to thank all of the study participants and their families, and all of the site investigators, study coordinators, and staff. Assistance in preparing this article for publication was provided by Genentech, Inc.

Alzheimer's Disease Neuroimaging Initiative (ADNI) is funded by the National Institute on Aging, the National Institute of Biomedical Imaging and Bioengineering, and through generous contributions from the following organizations: AbbVie; Alzheimer's Association; Alzheimer's Drug Discovery Foundation; Araclon Biotech; BioClinica, Inc.; Biogen; Bristol-Myers Squibb Company; CereSpir, Inc.; Cogstate; Eisai Inc.; Elan Pharmaceuticals, Inc.; Eli Lilly and Company; EuroImmun; F. Hoffmann-La Roche Ltd and its affiliated

company Genentech, Inc.; Fujirebio; GE Healthcare; IXICO Ltd.; Janssen Alzheimer Immunotherapy Research & Development, LLC.; Johnson & Johnson Pharmaceutical Research & Development LLC.; Lumosity; Lundbeck; Merck & Co., Inc.; Meso Scale Diagnostics, LLC; NeuroRx Research; Neurotrack Technologies; Novartis Pharmaceuticals Corporation; Pfizer Inc.; Piramal Imaging; Servier; Takeda Pharmaceutical Company; and Transition Therapeutics. The Canadian Institutes of Health Research is providing funds to support ADNI clinical sites in Canada. Private sector contributions are facilitated by the Foundation for the National Institutes of Health (www.fnih.org). The grantee organization is the Northern California Institute for Research and Education, and the study is coordinated by the Alzheimer's Therapeutic Research Institute at the University of Southern California. ADNI data are disseminated by the Laboratory for NeuroImaging at the University of Southern California.

The Australian Imaging Biomarkers and Lifestyle (AIBL) study (www.AIBL.csiro.au) is a consortium between Austin Health, CSIRO, Edith Cowan University, the Florey Institute (The University of Melbourne), and the National Aging Research Institute. Partial financial support was provided by the Alzheimer's Association (US), the Alzheimer's Drug Discovery Foundation, an anonymous foundation, the Science and Industry Endowment Fund, the Dementia Collaborative Research Centres, the Victorian Government's Operational Infrastructure Support program, the McCusker Alzheimer's Research Foundation, the National Health and Medical Research Council, and the Yulgilbar Foundation. Numerous commercial interactions have supported data collection and analysis. In-kind support has also been provided by Sir Charles Gairdner Hospital, Cogstate Ltd., Hollywood Private Hospital, the University of Melbourne, and St. Vincent's Hospital.

The BIOCARD study is supported by a grant from the National Institute on Aging : U19-AG03365. The BIOCARD Study consists of 7 Cores with the following members: (1) the Administrative Core (Marilyn Albert, Barbara Rodzon, Corinne Pettigrew), (2) the Clinical Core (Marilyn Albert, Anja Soldan, Rebecca Gottesman, Ned Sacktor, Scott Turner, Leonie Farrington, Maura Grega, Gay Rudow, Scott Rudow, Rostislav Brichko), (3) the Imaging Core (Michael Miller, Susumu Mori, Tilak Ratnanather, Timothy Brown, Hayan Chi, Anthony Kolasny, Kenichi Oishi, Laurent Younes), (4) the Biospecimen Core (Richard O'Brien, Abhay Moghekar, Jacqueline Darrow, Alexandria Lewis), (5) the Informatics Core (Roberta Scherer, David Shade, Ann Ervin, Jennifer Jones, Hamadou Coulibaly, April Broadnax, Lisa Lassiter), (6) the Biostatistics Core (Mei-Cheng Wang, Jiangxia Wang, Yuxin Zhu), and (7) the Neuropathology Core (Juan Troncoso, Olga Pletnikova, Gay Rudow, Karen Fisher). The authors would like to acknowledge the contributions to BIOCARD of the Geriatric Psychiatry Branch (GPB) of the intramural program of the National Institute of Mental Health who initiated the study (PI Dr. Trey Sunderland).

The BioFINDER study was supported by the European Research Council, the Swedish Research Council, the Strategic Research Programme in Neuroscience at Lund University (MultiPark), the Crafoord Foundation, the Swedish Brain Foundation, The Swedish Alzheimer foundation, the Torsten Söderberg Foundation at the Royal Swedish Academy of Sciences, and the regional agreement on medical training and clinical research (ALF) between Region Skåne and Lund University.

Data were provided in part by OASIS-3: Longitudinal Multimodal Neuroimaging: Principal Investigators: T. Benzinger, D. Marcus, J. Morris; NIH P30 AG066444, P50 AG00561, P30 NS09857781, P01 AG026276, P01 AG003991, R01 AG043434, UL1 TR000448, R01 EB009352. AV-45 doses were provided by Avid Radiopharmaceuticals, a wholly owned subsidiary of Eli Lilly.

The Harvard Aging Brain Study (HABS - P01AG036694; https://habs.mgh.harvard.edu) was launched in 2010, funded by the National Institute on Aging. and is led by principal investigators Reisa A. Sperling MD and Keith A. Johnson MD at Massachusetts General Hospital/Harvard Medical School in Boston, MA.

The Wisconsin Registry for Alzheimer's Prevention (WRAP) dataset were supported with grants from the US National Institutes of Health (grant Nos. AG027161 and AG021155).

The FACEHBI study was supported by funds from Fundació ACE Institut Català de Neurociències Aplicades, Grifols, Life Molecular Imaging, Araclon Biotech, Alkahest, Laboratorio de Análisis Echevarne and IrsiCaixa.

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

# A  Appendix

## A.1  Self-supervised Models

**SimCLR**
In this method, the learning rate is $1e - 4$, Adam is used as an optimizer and NT-Xent loss is used as a loss function. We use an implementation of SimCLR in Pytorch-lightning repository for creating our framework. `https://github.com/Lightning-AI/lightning-bolts.git`
**Barlow Twins**
This method utilizes LARS as an optimizer and $1e - 4$ as a learning rate scheduler, with CosineWarmup serving as the learning rate scheduler. This repository is used as a basis `https://github.com/SeanNaren/lightning-barlowtwins.git].`
**SwAV**
For this method, we use Adam as the optimizer, with a learning rate of $1e-4$. Like SimCLR, we use the SwAV base model implementation from the Pytorch-lightning repository `https://github.com/Lightning-AI/lightning-bolts.git`.

## A.2  MSE values corresponding to Table 4b

| Pretraining Method | Pretraining Dataset | MSE on in-study test | MSE on ABBY | MSE on OASIS-3 |
|---|---|---|---|---|
| Random | - | 5.88 | 6.11 | 4.03 |
| Barlow Twins | ImageNet | 5.49 | 5.89 | 4.11 |
| SimCLR | In-domain | 5.22 | 5.42 | 3.45 |
| Supervised ImageNet → SimCLR | Labeled_ImageNet → In-domain | 5.56 | 5.65 | 3.49 |
| Barlow Twins → SimCLR | Unlabeled_ImageNet → In-domain | 5.11 | 5.33 | 3.20 |

Table 6: Results of different pretraining schemes on test sets in terms of MSE.

