# OpenReview forum: "Robust Alzheimer's Progression Modeling using Cross-Domain Self-Supervised Deep Learning"
_TMLR — Accepted by TMLR_

### Review · Reviewer_6d7X · 2023-02-20

**Summary Of Contributions:**

The paper investigates different pre-training strategies for the prognosis of Alzheimer's disease progression using neural networks.
The investigated strategies include supervised/self-supervised pre-training with ImageNet followed by self-supervised pre-training on a large medical dataset and self-supervised pre-training on a large medical dataset w/ random initialization.
The results show that self-supervised pre-training with ImageNet followed by self-supervised pre-training on a large medical dataset led to the best scores on an unseen dataset.

**Audience:**

Yes

**Broader Impact Concerns:**

I don't have any broader impact concerns.

**Claims And Evidence:**

No

**Requested Changes:**

- Reporting results of multiple runs is crucial to understanding which method is better (please also see my first comment in the weaknesses section). Currently, it is difficult to reach conclusions reported in the paper.

- The information about the medical dataset split that are used during pre-training, fine-tuning, and test should be clarified. There should also be experiments with different test sets used; e.g. by rotating the test dataset with the ones used during pre-trainined and/or fine-tuning.

- I would be interested in seeing at least one experiment where the result of the fully supervised setting on the final test set is reported. This would give the idea about the gap between self-supervised pre-training and the upper bound.

**Strengths And Weaknesses:**

Strengths:
- Self-supervised pre-training is quite important in many applications, especially in the medical domain, where the number of labeled examples is limited. There are numerous self-supervised techniques and combinations of different pre-training strategies. Therefore, it is quite useful to investigate a subset of these methods and report the results for a certain application. In that sense, I think such an analysis would be of interest to many researchers in the field.
- Description of the analysis reported in the paper is explained clearly except for a few convoluted pieces of information that are crucial.

Weaknesses:
- The results reported in the paper after a single training run. Sometimes, the quantitative results are very close to each other and it might be difficult to conclude that one is better than the other without seeing the average+-std results of multiple runs with different seeds. Also, a statistical significance tests between these results would be useful to really conclude that the one method is better than the other.

- I didn't quite understand the medical datasets that are used in self-supervised pre-training, fine-tuning and the final evaluation. Are there any datasets that are used in both pre-training and final evaluation (I assume images do not overlap but I am asking if samples from one dataset can appear in different splits?) If so, this doesn't really show the generalization performance of the models. These splits should be indicated clearly.

- The results are obtained on a single test dataset. How would the results change if different test sets were used? I would like to see an experiment designed as follows to evaluate the results better: 1) medical datasets to be used for pre-training are fixed (let's say datasets A, B, C) and multiple datasets reserved for fine-tuning and final testing (let's call them dataset D, E), 2) model is pre-trained using A, B, C, 3) in one experiment, the pre-trained model is fine tuned on D and tested on E, and in the other vice versa. And, make sure that D and E are not very close to each other. I think this would be a more realistic setup to evaluate the performance. There is an experiment with an independent test set, but which dataset is not specified. And also, what are the test datasets used in the other experiments?

- What would be the upper bound when a model is trained on the target dataset with full supervision using sufficient labels? Can such an experiment be designed? Such an analysis would help to understand the gap between the upper bound and the self-supervised pre-training.

- Most of the results are reported in terms of R^2, but only a single experiment is reported with R^2 and Pearson's correlation. I think all results should be given with both metrics as it is difficult to interpret R^2.

Minor points:
- In Fig. 2, the results are given with different ranges in the axes of Figures a, b, and c. Currently, the difference seems very small by looking at the plots. Fixing the axes' ranges might help.
- On page 9 of the results section: I think Table 3a should be Table 3b.

---

> ### Author Response · Authors · 2023-06-09
> **First round of response to Reviewer 6d7X**
>
> $\star$ Reporting results of multiple runs is crucial to understanding which method is better (please also see my first comment in the weaknesses section). Currently, it is difficult to reach conclusions reported in the paper. In Fig. 2, the results are given with different ranges in the axes of Figures a, b, and c. Currently, the difference seems very small by looking at the plots. Fixing the axes' ranges might help.
> On page 9 of the results section: I think Table 3a should be Table 3b.
>
> Response: Definitely! To address this comment, we ran 3-fold cross-validation three times for all experiments and reported the average performance in terms of both $R^2$ and Pearson correlation ($r$) accompanied by their respective standard deviations. The results are presented in Table 4a. We also perform Steiger's Z1 method to find statistical difference between the resulting Pearson correlation values and show the results in Table 5 in revised paper. We also revised Figure 2 to present the frequency of the residuals  (the differences between the predicted CDR-SB and ground truth values) averaged over three-folds cross-validation setup. Figure 2 shows that our best performing model has the best overall goodness of fit, as its predicted CDR-SB values most closely match the ground truth CDR-SB values.
> We also fixed the typo in page 9. Thank you for bringing it to our attention.
> ________________
> $\star$ The information about the medical dataset split that are used during pre-training, fine-tuning, and test should be clarified. There should also be experiments with different test sets used; e.g. by rotating the test dataset with the ones used during pre-trainined and/or fine-tuning.
>
> Response: Sure! In response to your comments, we have thoroughly revised the 'Task and Data' section to improve its clarity and comprehension. We added Table 1 in our revised version to clearly indicate the splits. Our previous test set, which we now refer to as the in-study test set, included subjects from the pretraining dataset, but there was no overlap between the splits. To address your comment about the generalizability of our model, we have added two new datasets, OASIS-3 and study E, as out-of-study test sets. It is important to note that these two datsets are not utilized for either self-supervised pre-training or fine-tuning. Their purpose is to evaluate the generalizability of our model on external datasets. Therefore, we now have three test sets: two out-study test sets and one in-study test set (please see Table 1).
> _________________
> $\star$ I would be interested in seeing at least one experiment where the result of the fully supervised setting on the final test set is reported. This would give the idea about the gap between self-supervised pre-training and the upper bound.
>
> Response: Certainly! The initial row in Table 4a displays the outcomes of the Fully Supervised setting, where no SSL pretraining was conducted, and the ResNet-50 backbone was randomly initialized. Similary Table 4b top row shows the outcome of the fully supervised setting on different test set.

---

### Review · Reviewer_cwKK · 2023-05-08

**Summary Of Contributions:**

This research paper explores the potential of using self-supervised learning (SSL) techniques to assess the progression of Alzheimer's disease. Specifically, it utilises various SSL approaches such as SimCLR, BarlowTwins, and SWaV to evaluate the effectiveness of SSL methods in a medical setting.

The paper is noteworthy as it is the first to consider multiple SSL methods and evaluate how different data combinations, such as pretraining on imagenet/other medical datasets and then using Alzheimer's as the downstream task, can be used to assess Alzheimer's progression. Additionally, the paper highlights SSL's usefulness in predicting a score related to Alzheimer's progression more accurately.

It should be noted that there is no technical contribution in terms of self-supervised learning, but it might appeal to the medical community as a means to apply SSL in other settings.



**Audience:**

No

**Broader Impact Concerns:**

there is no reflection regarding clinical evaluation of any sort, or how such a system could be used in a more realistic setting. The ML community will not know how the clinically relevant score to evaluate Alzheimer's progression is used in practice.

Could you please confirm that the internal datasets used correspond to a clinical study that has received the relevant ethics approvals?

**Claims And Evidence:**

No

**Requested Changes:**

--The results do not have to be scattered like that but instead, a table should summarise the findings, across SSL methods and combinations of data.

-- Rather than having speculative headings for the sub sections in section 4, please use informative headings and then discuss the findings and limitations when showing the results.



**Strengths And Weaknesses:**

Strengths

-- large-scale evaluation using three main SSL methods
-- empirical results on utilising ImageNet and other Medical Imaging Datasets to pre-train SSL for the main downstream task
-- promising results that could be transferred to other medical conditions.

Weaknesses

-- overly non-technical; the technical bit reads more like a blog or guidelines
-- R^2 is uninformative as a means to provide a quantitative evaluation of deep learning model fit. It is a metric suitable for linear models mostly to explain variance when manipulating various independent variables.
-- Not very well written, e.g. what is the purpose of the sub-section 4.2 heading?
-- What is the takeaway message of figure 2? For instance, are you trying to fit a linear model on the predicted vs ground truth data?

---

> ### Author Response · Authors · 2023-06-09
> **First round of response to Reviewer cwKK**
>
> $\star$ The results do not have to be scattered like that but instead, a table should summarise the findings, across SSL methods and combinations of data.
>
> Response: Sure! To address your comment, we completely rewrote the 'Methodology' and 'Results' section to enhance its clarity and readability. We gathered all of the results in table 4-a and table 4-b which present the performance of best performing models on validation set, in-study test set and out-study test sets. We also reported the statistical difference between the resulting Pearson correlation values ($r$) in Table 5 in revised paper.
> _____________________
> $\star$ Rather than having speculative headings for the sub sections in section 4, please use informative headings and then discuss the findings and limitations when showing the results.
>
> Response: We revised the  'Results' section, enhancing it with more informative headings and providing detailed discussions of the results in each respective section. We modified Table 4 to report the results using both the coefficient of determination ($R^2$) and the Pearson correlation ($r$). We also calculated the statistical difference between the resulting Pearson correlation values and reported the results in Table 5 in revised paper.
> Apologies for the confusion in Fig. 2. Based on your feedback, we revised Figure 2 to present the frequency of the residuals  (the differences between the predicted CDR-SB and ground truth values) averaged over three-folds cross-validation setup. Figure 2 shows that our best performing model has the best overall goodness of fit, as its predicted CDR-SB values most closely match the ground truth CDR-SB values.
> _______________________
> $\star$ there is no reflection regarding clinical evaluation of any sort, or how such a system could be used in a more realistic setting. The ML community will not know how the clinically relevant score to evaluate Alzheimer's progression is used in practice.
>
> Response: Sorry for the confusion. We totally changed our 'Methodology' section to indicate that the CDR SB is a score provided by clinicians based on clinical evaluations. CDR-SB score is then used to assign Alzheimer's status of a patient. In our study, we show that our model correlates with the CDR-SB score and predicts AD progression. By accurately predicting the progression to AD in patients, clinicians can initiate treatment at an earlier stage and tailor the most suitable and effective treatment for each individual patient.
> _______________________
> $\star$ Could you please confirm that the internal datasets used correspond to a clinical study that has received the relevant ethics approvals?
>
> Response: Yes, the internal datasets used in the study correspond to a clinical study that has received the relevant ethics approvals.
> ______________________

---

> > ### Comment · Reviewer_cwKK · 2023-06-16
> > **response**
> >
> > Many thanks for the response - I think one of the other reviewers has also highlighted the need to include accuracy for classification tasks and MSE for regression tasks in your experiments. I do not see either of them to have been included

---

> > > ### Author Response · Authors · 2023-06-17
> > > **Including MSE scores**
> > >
> > > Thank you for your feedback. Regarding other reviewer's comment, we have thoroughly revised the "Task and Data" section to provide a clear and explicit definition of our task, that is a regression task (we do not discuss any classification task in the paper).  Accordingly, we have also incorporated MSE scores in Table 4a in the revised version. To avoid overcrowding Table 4b with too many numbers, we have moved the MSE scores for the models in Table 4b to Table 6 in the supplementary material.

---

> > > > ### Comment · Reviewer_cwKK · 2023-06-19
> > > > **final comment**
> > > >
> > > > Thanks for making the additional changes - I am happy with the changes. One thing to highlight is that I personally do not think that R^2 is an appropriate measure of fit for the types of models we are dealing with here, but MSE (and MAE and RMSE) are more representative of the predictive nature of the approaches. I will finalise my review and submit my recommendation.

---

### Review · Reviewer_XPWA · 2023-05-29

**Summary Of Contributions:**

This paper proposed a self-supervised learning scheme for brain MR image analysis in AD patients with the help of networks trained on natural images. The model was evaluated with ablation studies on the task of AD classification.

**Audience:**

Yes

**Broader Impact Concerns:**

This work involves little if any ethical concerns.

**Claims And Evidence:**

Yes

**Requested Changes:**

See my comments above.

**Strengths And Weaknesses:**

Strength:
1) Trained and evaluated on a large, multi-institutional dataset.
2) Investigated multiple settings for self-supervised learning and the schemes for utilizing natural image pre-trained model.

Weakness:
The "Methodology" section and Fig. 1 need substantial improvement to understand the experiment setting, specifically:
1) What is the term "defined comparison" in Fig. 1? The term has never been described in the manuscript. This is critical as self-supervised learning relies on a certain invariance or intrinsic data structure. But the reviewer failed to understand what pairs of MR images are defined as "similar" and not.
2) It is unclear what exact task has been performed in the experiment, although it is likely a classification task. In addition, while the author claimed that R2 is a better metric, it is suggested that accuracy be reported so that the reader can compare the model's performance with other methods.

---

> ### Author Response · Authors · 2023-06-09
> **First round of response to Reviewer XPWA**
>
> $\star$ The "Methodology" section and Fig. 1 need substantial improvement to understand the experiment setting, specifically  What is the term "defined comparison" in Fig. 1? The term has never been described in the manuscript. This is critical as self-supervised learning relies on a certain invariance or intrinsic data structure. But the reviewer failed to understand what pairs of MR images are defined as "similar" and not.
>
> Response: Sure! to address your comment, we completely rewrote the 'Methodology' section to enhance its clarity and readability. We have redesigned Figure 1 and revised its caption accordingly. It is important to note that our task is specifically a regression problem, as we have clarified in the revised version. We also provided clear details regarding the composition of the training and test sets. We added Table 1 in our revised version to clearly indicate the splits. Furthermore, we included the Pearson correlation as an additional evaluation metric (please see Table 4).
> _______________________
> $\star$ It is unclear what exact task has been performed in the experiment, although it is likely a classification task.
>
> Response: Thank you very for your feedback. We have taken your comment into consideration and have thoroughly revised the 'Task and Data' section to provide a clear and explicit definition of our task which is a regression task.
> ______________________
> $\star$ In addition, while the author claimed that R2 is a better metric, it is suggested that accuracy be reported so that the reader can compare the model's performance with other methods.
>
> Response: Right! As our task involves regression, we also added the Pearson correlation ($r$) as an additional evaluation metric (please see Table 4). We also calculated the statistical difference between the resulting Pearson correlation values and reported the results in Table 5 in revised paper. Furthermore, we introduced two distinct test sets, referred to as the out-study test sets, which were not utilized for either semi-supervised learning or fine-tuning (please see table 1). These test sets allow us to assess the generalization of our model to external datasets. We also revised Figure 2 to present the frequency of the residuals  (the differences between the predicted CDR-SB and groundtruth values) averaged over three-folds cross-validation setup. Figure 2 shows that our best performing model has the best overall goodness of fit, as its predicted CDR-SB values most closely match the ground truth CDR-SB values.

---

### Decision · Action_Editors · 2023-07-01

**Recommendation:** Accept as is

**Comment:**

This paper analyzes the impact of pre-training on the task of Alzheimer's disease progression. The authors investigate the use of supervised and self-supervised learning strategies, especially SimCLR, BarlowTwins, and SWaV.

In their initial reviews, the reviewers found the experimental study interesting but pointed out the lack of methodological contributions in the work. They also raised concerns about the evaluation protocol (dataset splits), the absence of statistical significance testing, and the need to use more relevant metrics beyond $R^2$. RcwKK and RXPWA additionally highlighted some limitations in the paper presentation. The revision overall successfully improved the experiments, clarified the results, provided consolidated findings, and highlighted the link between the results and the clinical outcome. After the discussion period, R6d7X and RcwKK recommended weak acceptance due to the limited novelty and the interesting experiments, while RXPWA was more inclined to reject the paper.

The AE carefully reviewed the submission and discussions. The AC agrees that there is no methodological contribution, but also considers that the experimental results are relevant. The main finding of this work is that self-supervised pre-training on ImageNet, followed by self-supervised pre-training on a large medical dataset, yielded the best results for estimating Alzheimer's disease progression. While it would have been beneficial to consider other recent baselines, such as foundation models (e.g. CLIP), the paper's claims are supported by the experiments and would certainly be of interest to a non-negligible subset of the TMLR audience. Therefore, the AE recommends accepting the paper.


**Audience:**

The paper's results would certainly be of interest to a non-negligible subset of the TMLR audience

**Claims And Evidence:**

Yes